# A 3D Model of Human Buccal Mucosa for Compatibility Testing of Mouth Rinsing Solutions

**DOI:** 10.3390/pharmaceutics15030721

**Published:** 2023-02-21

**Authors:** Paula Zwicker, Maxi Zumpe, Axel Kramer, Gerald Müller

**Affiliations:** 1Institute of Hygiene and Environmental Medicine, University Medicine, Ferdinand-Sauerbruch-Straße, D-17475 Greifswald, Germany; 2Department of Pediatric Hematology and Oncology, University Medicine, Ferdinand-Sauerbruch-Straße, D-17475 Greifswald, Germany

**Keywords:** mucosa model, mouth rinse, mucositis, TEER

## Abstract

Oral mucositis is the most common and severe non-hematological complication associated with cancer radiotherapy, chemotherapy, or their combination. Treatment of oral mucositis focuses on pain management and the use of natural anti-inflammatory, sometimes weakly antiseptic mouth rinses in combination with optimal oral cavity hygiene. To prevent negative effects of rinsing, accurate testing of oral care products is necessary. Due to their ability to mimic realistic in-vivo conditions, 3D models may be an appropriate option in compatibility testing of anti-inflammatory and antiseptically effective mouth rinses. We present a 3D model of oral mucosa based on the cell line TR-146 with a physical barrier, characterized by high transepithelial electrical resistance (TEER) and confirmed cell integrity. Histological characterization of the 3D mucosa model showed a stratified, non-keratinized multilayer of epithelial cells similar to that of human oral mucosa. By means of immuno-staining, tissue-specific expression of cytokeratin 13 and 14 was shown. Incubation of the 3D mucosa model with the rinses had no effects on cell viability, but TEER decreased 24h after incubation in all solutions except ProntOral^®^. Analogous to skin models, the established 3D model meets the quality control criteria of OECD guidelines and may therefore be suitable for comparing the cytocompatibility of oral rinses.

## 1. Introduction

Mucositis is the most common and most severe non-haematological complication of cancer chemo- or radiotherapy [1]. The treatment is focused on pain management and the use of natural anti-inflammatory and sometimes weakly antiseptic compounds in combination with optimal oral cavity hygiene. National guidelines recommend rinsing with water or 0.9% saline solution as an additional measure [2]. Due to limited data, there are no rinses recommended as alternatives to saline solution in cases of oral mucositis. To prevent the negative effects of rinsing, sufficiently sensitive testing of oral care products and their mucosal cytocompatibility is necessary.

However, most in vitro studies [3,4,5,6,7,8,9] have limited significance, because they use cell lines established from mouse or simple monolayer cell cultures of cell lines or isolated cells. Tissue cultures of explants of neonatal rat peritoneum are better suited for testing the compatibility of mouth rinses [10].

Validated 3D human epidermal models have been successfully introduced by OECD guidelines as in vitro alternative models for testing skin or eye irritation [11], skin sensitization [12], and skin corrosion [13]. Since 3D models mimic more realistic in-vivo conditions with their physiologically tissue-like structure grown in culture, they may also be an appropriate option in compatibility testing of antiseptics and mouth rinses. Unfortunately, there are no guidelines for using oral mucosa equivalents for such applications.

Even if tissue-engineered oral mucosa equivalents have been used in scientific research for investigating Candida–host interactions [14,15], the biology of cancer [16,17], the metabolic impact of xenobiotics [18,19], biofilm formation [20], drug delivery [21,22,23], as well as clinical applications [24,25], studies utilizing them for tissue compatibility testing are rare [7,26,27,28].

For these test setups, TR-146 cells seem to be a preferred cell line for generating a reproducible 3D model to investigate tissue compatibility, as it is commercially available as HOE (human oral epithelium) from SkinEthic Laboratories (Nice, France) or as EpiOral™ from MatTek Corporation (Ashland, MA, USA). With these cells, this study establishes a model of oral mucosa characterized by high transepithelial electrical resistance (TEER) and histologically similar to human oral mucosa. The cytocompatibility of several commercially available antiseptic mouth rinses was investigated using the presented 3D human oral mucosa model.

## 2. Materials and Methods

### 2.1. Cell Culture

TR-146 cells, a cell line established from a human squamous cell carcinoma, were cultured in DMEM/F12 cell culture medium (PAN-Biotech GmbH, Aidenbach, Germany) with the addition of 2 mM glutamine (PAN-Biotech GmbH, Aidenbach, Germany) and 10% fetal bovine serum (FBS, Gibco Thermo Fisher Scientific, Waltham, MA, USA) in T75 tissue culture flasks at 37 °C in a humidified atmosphere (5% CO_2_). For cultivation, cells were detached at a confluency of about 80% using 0.05% trypsin/0.02 mM EDTA after rinsing the cells with PBS without Ca^2+^/Mg^2+^ (woPBS, ccPro, Vogtei, Germany). Morphology was checked regularly. The cells were free of mycoplasma. Cell viability was ensured by trypan blue exclusion.

TR-146 cells are non-differentiated and non-keratinized stratified epithelial cells which present characteristics similar to normal buccal mucosa [29].

### 2.2. 3D Cell Culture

In a 6-well cell culture plate, one cell culture insert (MilliCell 0.4 mm PCF, 12 mm Diameter, Merck Millipore, Burlington, MA, USA) was placed in each well and 500 µL Medium A were added into the well. Medium A was composed of 485 mL DermaLife^®^ basal medium (CellSystems GmbH, Troisdorf, Germany) with the addition of 5 mL human keratinocyte growth supplement (Life Technologies, Carlsbad, CA, USA), 0.01 mg/mL gentamicin (GE Healthcare, Chicago, IL, USA), and 0.25 µg/mL amphotericin B (PAN-Biotech GmbH, Aidenbach, Germany). A cell suspension was adjusted to a cell concentration of 1 × 10^6^ cells/mL in medium A, and into each insert, 300 µL of this suspension were added. After incubation for 24 h, 1 mL medium A was added to each well, the medium in the inserts was removed, and 500 µL fresh medium were added. The next day, the medium in the well and the insert was renewed, and incubation was continued for 24 h. Then, medium was removed from the inserts and the well. The inserts were exposed to the air for 10 min in a laminar air-flow chamber under sterile conditions to enable cell differentiation (day 0). Afterwards, the inserts were placed back into the 6-well cell culture plates and 1.5 mL medium B were added. Medium B consists of medium A plus 0.9 mM CaCl_2_ for cell differentiation. Subsequently, medium in the well was renewed every other day. On day 14, the inserts were ready for further use.

### 2.3. Transepithelial Electrical Resistance Measurement (TEER)

The inserts were placed in 24-well cell culture plates filled with 600 µL PBS (with Ca^2+^/Mg^2+^; wPBS) per well and washed by adding 400 µL wPBS following vacuum aspiration. A 400 µL volume of fresh wPBS was added and the TEER was measured to ensure integrity of the artificially obtained cell complex using the EVOM™ Epithelial Volt-ohmmeter with the 5TX2 electrode (World Precision Instruments Ltd., Sarasota, FL, USA). An insert without cells was used as a blank.

### 2.4. Fixation, Embedding, and Cryo-Sectioning

Fixation of cells was carried out on day 14. The inserts were transferred to a 24-well cell culture plate containing 2 mL HistoChoice^®^ Tissue Fixative (Molecular Biology, Electron Microscopy Sciences, Hatfield, PA, USA). After fixation for 2 h at room temperature, the inserts were washed twice with 2 mL woPBS followed by embedding in 20% saccharose solution for 24 h at 4 °C.

The 3D cell layer was disassociated from the insert and embedded in Tissue-Tek^®^ O.C.T. ™ compound (Sakura Finetek Europe, Alphen aan den Rijn, The Netherlands) on the frozen cryostat (CM1900, Leica Biosystems GmbH, Wetzlar, Germany) before cutting slices of approximately 5–6 µm thickness. The cryo-sections were transferred to a microscope slide at room temperature and dried at 45 °C for 18 h.

### 2.5. HE-Staining of Cryo-Sections

After 2 min rehydration in distilled water, tissue sections were incubated in Mayer’s Hemalum stain (Carl Roth GmbH + Co. KG, Karlsruhe, Germany) for 4 min following rinsing for 10 min with tap water. After further staining for 5 min in eosin G (Carl Roth GmbH + Co. KG, Karlsruhe, Germany), the cryo-sections were briefly dipped in 70% ethanol. Next, the cryo-sections were dehydrated in ascending concentrations of ethanol. In the last step, they were dipped in xylene and covered with DPX mountant (Sigma Aldrich, St. Louis, MO, USA).

### 2.6. Immunohistochemical Detection of Cytokeratines CK-13 and CK-14

Cryo-sections were incubated in Tris-buffered saline (TBS) for 10 min at 4 °C and rehydrated [30,31]. Endogenous peroxidase was blocked by addition of 1% H_2_O_2_/TBS for 30 min [30,31]. Then, the cryo-sections were blocked in 3% goat serum (Sigma Aldrich, St. Louis, MO, USA)/TBS for 30 min at room temperature followed by incubation with mouse anti-human cytokeratine-13-IgG (0.1 mg/mL, antibodies-online GmbH, Aachen, Germany) or mouse anti-human cytokeratine-14-IgG antibodies (0.04 mg/mL; antibodies-online GmbH, Aachen, Germany) for a maximum of 24 h at 4 °C in a humidified atmosphere. Afterwards, the cryo-sections were incubated with biotin-conjugated goat anti-mouse antibody (IgG-Fc, Life Technologies GmbH, Darmstadt, Germany) for 30 min at room temperature in a humidified atmosphere with subsequent incubation in an avidin-peroxidase solution (Invitrogen AG, Carlsbad, CA, USA) for 30 min. The bound peroxidase was visualized by using 3,3′-diaminobenzidine (Sigma Aldrich, St. Louis, Missouri, USA). After 15 min incubation, the cryo-sections were washed with TBS and covered with Mowiol 4–88 (Carl Roth GmbH + Co. KG, Karlsruhe, Germany). Washing followed every incubation step.

### 2.7. Cell Viability Testing

Cell viability of the 3D models 24 h after treatment was analyzed using thiazolyl blue tetrazolium bromide (MTT, Sigma Aldrich, St. Louis, MO, USA). Per well, 500 µL of MTT medium (cell culture medium DMEM/F12 with glutamine and FCS, 0.5 mg/mL MTT) were added and inserts were inserted. After incubation for 3 h, the inserts were transferred to 2 mL elution solution consisting of 96 mL propan-2-ol and 4 mL 1 M HCl. Inserts were incubated overnight at 4 °C. Afterwards, 200 µL were transferred to a 96-well plate and absorbance was measured at 560 nm.

### 2.8. Quality Control

For testing the barrier function, the 3D mucosa models were treated with Triton X-100 (1%, Sigma Aldrich, St. Louis, Missouri, USA) for several incubation periods (0–5 h) as well as with SDS (AppliChem GmbH, Darmstadt, Germany) for 18 h at different concentrations (0–2 mg/mL), according to tests used for epiCS^®^ and KeraSkin™SIT epidermis models as given in OECD test guideline no. 439. The cells were treated with 30 µL (Triton X-100) or 40 µL (SDS) solution.

After treatment, viability was analyzed using MTT assay.

### 2.9. Treatment of 3D Models with Test Solutions and Cytokine Analysis

First, the effect of single antimicrobial agents on TEER values was tested by adding equimolar concentrations (1.6 mM) of poly (hexamethylene biguanide) hydrochloride (PHMB, 0.45%), chlorhexidine digluconate (CHG, 0.14%), or octenidine dihydrochloride (OCT 0.1%) for 5 min to the 3D models in a volume of 50 µL. TEER was measured directly after treatment.

Second, the cell models were treated with oral rinses for 30 s to mimic realistic conditions. Furthermore, Triton X-100 (1%, 2%), SDS (0.5%, 1%), and LPS from *E. coli* O55:B5 (50 µg/mL, Sigma Aldrich, St. Louis, MO, USA) were used as positive controls. PBS served as the negative control.

The inserts were washed once with PBS; 30 s after equilibration with PBS, the TEER was measured. PBS was removed and 50 µL of mouth rinse or control solutions were added. After exposition for 30 s, cells were washed five times with 400 µL PBS, and TEER was determined again. The inserts were placed in fresh cell culture medium (1 mL) and further incubated for 24 h. Following this, supernatant was used for cytokine analysis, TEER was measured, and cell viability was analyzed by MTT assay. Until use, the supernatant was stored at −80 °C. Cytokine secretion was quantified using ABTS ELISA Development kits (PeproTech EC Ltd., Cranbury, NJ, USA).

### 2.10. Test Products

For testing cytocompatibility in the 3D mucosa model, three antiseptic agents and five commercially available rinses were selected. The antiseptic agents PHMB (0.45%), CHG (0.14%) and OCT (0.1%), were dissolved in Ampuwa water (Fresenius Kabi, Bad Homburg vor der Höhe, Germany) at equimolar concentrations. Three commercial antiseptic mouth rinses based on these agents, and the two further rinses (Granudacyn^®^, Meridol^®^) were tested:−ProntOral^®^, containing 0.15% PHMB (B.Braun Melsungen AG, Melsungen, Germany)−Octenisept^®^, containing 0.1% OCT and 2% phenoxyethanol (Schülke & Mayr GmbH, Norderstedt, Germany)−Chlorhexamed^®^ FORTE without alcohol, containing 0.2% CHG, (GlaxoSmithKline Consumer Healthcare GmbH & Co. KG, München, Germany)−Granudacyn^®^, containing 50 ppm hypochlorite and 50 ppm hypochlorous acid (Mölnlycke^®^ Health Care GmbH, Düsseldorf, Germany)−Meridol^®^, containing amino fluoride and stannous fluoride (CP GABA GmbH, Hamburg, Germany).

### 2.11. Statistical Analysis

Data were analyzed using one-way ANOVA following Tukey’s multiple comparison test (TEER measurement) and Dunnett’s multiple comparison (MTT, cytokine analysis).

## 3. Results

### 3.1. Characterization of the 3D Models

After 14 d in air-lift culture, an uncornified squamous epithelium had grown (Figure 1). Basal cells developed a cubic form and cell nuclei were visible. The stratum intermedium comprised polygonal and cubic cells with cell nuclei. In contrast, the cells of the stratum superficiale were flattened, the cell borders were not clearly visible and cell nuclei were difficult to identify. The thickness of the whole model was about 150 µm consisting of 8–12 single cell layers.

Immunohistochemical staining of the cryo-sections to detect CK-13 and CK-14 was performed using a reaction of peroxidase with 3,3′-diaminobenzidine (Figure 2). CK-14 is a marker for stratified epithelial cells; CK-13 is mainly expressed in uncornified epithelial cells. CK-13 was found in the suprabasale layer of the cryo-sections, whereas CK-14 was mainly detected in the basal cells.

### 3.2. Quality Control

Mean TEER values were 388.8 ± 54.5 Ω (*n* = 107). The TEER values displayed an intra-lot coefficient of variation of 8.8 ± 4.7% (*n* = 12). Inter-lot, the coefficient of variation was about 14.2 ± 7.0% (*n* = 10). The cell viability assessed by MTT assay varied by 6.8% (*n* = 7).

The treatment with 1% Triton X-100 for 0 to 5 h led to a time-dependent loss of viability. The ET_50_ value was calculated to be 1.08–2.55 h. Treatment with SDS solution (0–2 mg/mL) for 18 h similarly induced a cell viability loss, depending on SDS concentration. The calculated IC_50_ value was about 1.8 mg/mL (0.18%).

### 3.3. TEER and Cell Viability after Treatment with Antiseptic Agents and Antiseptic Rinses

The TEER was measured directly before and after application of the antimicrobial agents for 5 min. In contrast to PHMB (0.45%), exposure of the 3D mucosa models to OCT (0.1%) and CHG (0.14%) led to a significant reduction of TEER values directly after treatment (Figure 3). TEER decreased to 70 ± 11.8% after OCT treatment and to 55 ± 13.0% after CHG exposure. Contact with PHMB reduced TEER only slightly, to 88 ± 9.1%; however, this decrease was not statistically significant. Contact to PBS did not affect TEER.

Directly after treatment with the commercial rinses, TEER values did not vary statistically significantly in comparison to the PBS-treated controls. TEER was significantly lower for Meridol^®^, Granudacyn^®^, Octenisept^®^, and Chlorhexamed^®^ 24 h after treatment in comparison to the PBS-treated control (Figure 4). The positive controls, Triton X-100 (2%) and SDS (0.5%, 1%), showed decreased cell viability as well. Treatment with LPS (50 µg/mL) did not reduce cell viability after 24 h in comparison to the PBS control.

The cell viability of the 3D mucosa models was not affected 24 h after the 30-s treatment with the rinses or the positive controls (Figure 5). Contact with Triton X-100 (2%) resulted in a significant reduction of cell viability.

### 3.4. Cytokine Secretion

Secretion of the cytokines TNFα, interleukin (IL)-1α, IL-6, and IL-8 was analyzed in the supernatant of the mucosa models 24 h after a 30-s treatment with antiseptic rinses using ELISA to test whether mouth rinses induced inflammation. Secretion of TNFα was not affected by the treatment of the mucosa models with the antiseptic rinses and positive controls (data not shown). In contrast, the concentration of IL-8 significantly increased after contact with ProntOral^®^ and SDS (1%). SDS (0.5%), LPS (50 µg/mL), and Meridol^®^ led to a statistically non-significant increase (Figure 6).

IL-1α secretion was slightly higher for mouth rinses than for the positive controls, but was not significantly elevated in comparison to the PBS control. The PHMB-based and the CHG-based products most strongly increased the IL-1α secretion (Figure 6).

IL-6 secretion was comparable for all tested mouth rinses and positive controls (Figure 6).

## 4. Discussion

Cell models mimicking natural mucosa are important for studying the effects of gargling solutions or antiseptics to ensure safety for users. Therefore, a mucosa model was established using TR-146 human squamous carcinoma cells. In 1995, Jacobsen et al. had already set up a 3D mucosa model using TR-146 cells cultivated on PET filter membranes, reaching the maximum cell integrity after 30 d, while being supplied with medium from apical and basolateral. Four to seven cell layers, organized as a stratified epithelium with clearly distinct surface cells, resulted from this cultivation procedure [32], representing the characteristics of a normal buccal epithelium. Later, Jacobsen et al. repeated their experiments, first submerging the culture in medium, and then supplying it basolaterally. However, the method did not lead to a higher TEER value and the permeability was higher in these models [33]. Further optimization was performed by Lin et al. to study transport processes through the epithelium. The highest TEER was measured at day 43 for the submerged culture, with a value of 61.94 ± 0.85 Ω [34]. The airlift culture had a TEER of about 158.14 ± 9.34 Ω. Data of the present study revealed that within 17 days, a 3D cell model similar to normal buccal mucosa was established, exhibiting a high TEER of 388.8 ± 54.5 Ω. Thus, the cell structure can be considered as tightly connected [35], resulting in a shortened cultivation time until mucosa models can be used.

Lin et al. revealed that both CK-14, a marker for stratified epithelial cells, and CK-13 were abundant in their 3D mucosa model [35,36,37]. In our study, after 14 days of airlift culture, an uncornified squamous epithelium with distinct cell layers had already grown. CK-13 and CK-14 were abundant suprabasally and basally, respectively; thus, our model represents a non-cornified stratified epithelium.

The OECD Guideline 439, outlining the conditions for skin models, states that models should resist SDS or Triton X-100 treatment, demonstrating the barrier function of the models. The data generated in our study are within the given ranges, as the ET_50_ of Triton X-100 (1%) was about 1.08–2.55 h and the IC_50_ for SDS was about 1.8 mg/mL. Klausner et al. [25] revealed an ET_50_ of 1.02 ± 0.33 h for their buccal mucosa model, which is comparable to our data. Furthermore, the viability should not vary by more than 18% between several models. We calculated a coefficient of variation of 6.8% by MTT assay. Our data furthermore reveal a low intra- and inter-lot coefficient of variation of TEER, confirming good reproducibility. Since the in-house 3D mucosa model of TR-146 cells met the quality control criteria given for skin models, it may be suitable for use in compatibility testing and comparable to commercially available mucosa models.

Thus, we used the model to test the antiseptics PHMB, OCT, and CHG (often used in mouth-rinses) in equimolar concentrations. Indeed, TEER values were significantly lower for the antiseptic agents CHG and OCT, suggesting possible negative effects to the mucosa if used often.

Afterwards, mucosa models were treated with commercially available oral and wound rinses for 30 s to mimic realistic conditions. After 24 h following contact with the oral rinses, no effects on cell viability were visible; however, TEER values were significantly reduced for all oral rinses except ProntOral^®^, which showed the most favorable performance. Regarding the release of IL-1α and IL-6, the antiseptic solutions did not differ significantly. In contrast, only ProntOral^®^ increased the release of IL-8 significantly. Increased IL-8 secretion may suggest an enhanced recruitment of neutrophil granulocytes, with increased phagocytosis of bacteria and destroyed tissue. However, elevated concentrations may also lead to tissue degradation [38] and increased inflammation, which would be counterproductive in cases of mucositis. None of the other test products showed a potential for increasing inflammation.

Even if CHG is often considered as the gold standard for oral antisepsis in the healthy oral cavity, a certain degree of cytotoxicity has been proven [39,40,41]. This cytotoxicity is probably responsible for the CHG-based rinses not significantly reducing the incidence or the severity of mucositis [42]. In fact, the opposite has been observed: patients treated with a CHG-based product have more problems with inflammation of the oral mucous membranes, resulting in an elevated mucositis score and an increase in C-reactive protein [43]. In vitro and in vivo studies of IL-1 secretion after CHG treatment provide very heterogenous results [4,44,45], which makes comparison very difficult. Together with data from the current study, the use of CHG-based oral rinses is not recommended for damaged mucosa, as is found in radio-or chemo-induced oral mucositis. Because of its weak antiseptic efficacy in vitro [46,47] and the statistically significant decrease of the TEER, Meridol^®^ is also not recommended for use in mucositis patients. Since cell integrity in mucositis patients is already threatened, a potentially suitable rinsing solution should not have an effect on TEER in vitro. That is why Granudacyn^®^ and Octenisept^®^ should also not be recommended, even if their impact on TEER is low, as shown by the present data.

Our study had some limitations. First, because we did not examine the permeability of the models, we cannot draw conclusions regarding the suitability for drug delivery testing, nor do we have any data regarding delivery of antiseptic agents to deeper cell layers. Furthermore, our study lacked a comparison to human buccal mucosa, since we did not examine our tests on excised mucosa. Finally, owing to the fact that we intended to establish a reproducible model, we used a cancer cell line that has a distinct phenotype in comparison to healthy cells, especially affecting wound healing and viability due to strengthened cell growth and reproduction. Because the 3D model closely resembles the situation of the buccal mucosa, it would be worthwhile to investigate its suitability for drug delivery testing.

## 5. Conclusions

The production of 3D mucosa models mimicking natural mucosa was established. The models met the quality control criteria of OECD guidelines valid for skin models and were thus used to test oral rinses for their cytocompatibility when used on oral mucosa.

To summarize, ProntOral^®^ appears to have the best tolerability of the antiseptic solutions tested regarding cell viability and TEER. Effects of increased IL-8 secretions must be analyzed further.

## Figures and Tables

**Figure 1 pharmaceutics-15-00721-f001:**
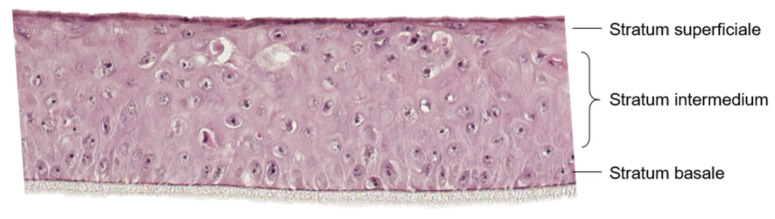
Representative image of a HE-stained, 5-µm cryo-section of the 3D mucosa equivalent, showing the three characteristic cell layers of the uncornified squamous epithelium with a thickness of about 150 µm on a porous membrane (10 µm), 200× magnification.

**Figure 2 pharmaceutics-15-00721-f002:**
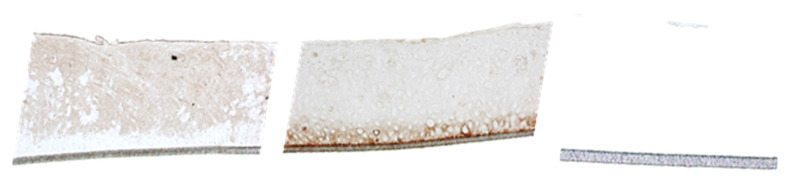
Verification of cytokeratin-13 in the suprabasale cell layer (**left**) and cytokeratin-14 in the basal cells (**middle**) and negative control (**right**). 200× magnification.

**Figure 3 pharmaceutics-15-00721-f003:**
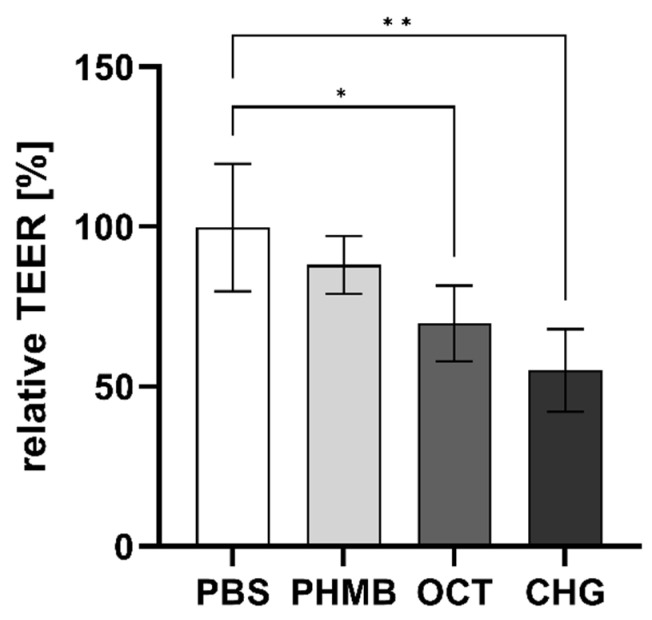
TEER (mean ± SD) after 5 min contact with poly (hexamethylene biguanide) hydrochloride (PHMB, 0.45%), chlorhexidine digluconate (CHG, 0.14%), and octenidine dihydrochloride (OCT, 0.1%) relative to TEER before treatment. For statistical analysis, ANOVA with the appropriate post-hoc test (Tukey’s multiple comparison test) was used. (each *n* = 4, * *p* < 0.05, ** *p* < 0.01 vs. PBS-treated control).

**Figure 4 pharmaceutics-15-00721-f004:**
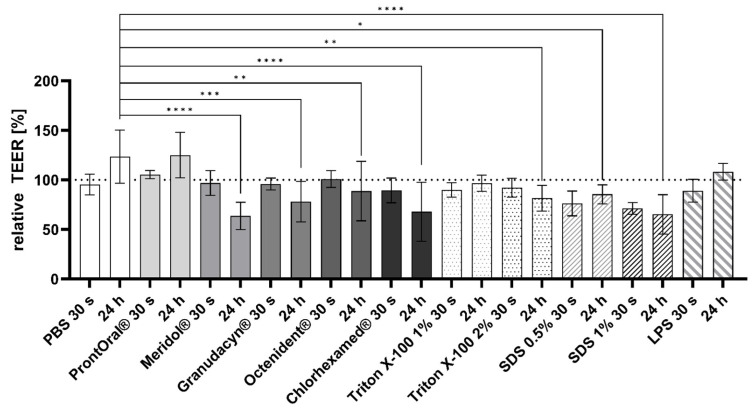
TEER (mean ± SD) measurement directly after 30-s treatment of the mucosa equivalents with the rinsing solutions, and 24 h after treatment (each *n* = 4–16, * *p* < 0.05, ** *p* < 0.01, *** *p* < 0.001, **** *p* < 0.0001 vs. PBS-treated control).

**Figure 5 pharmaceutics-15-00721-f005:**
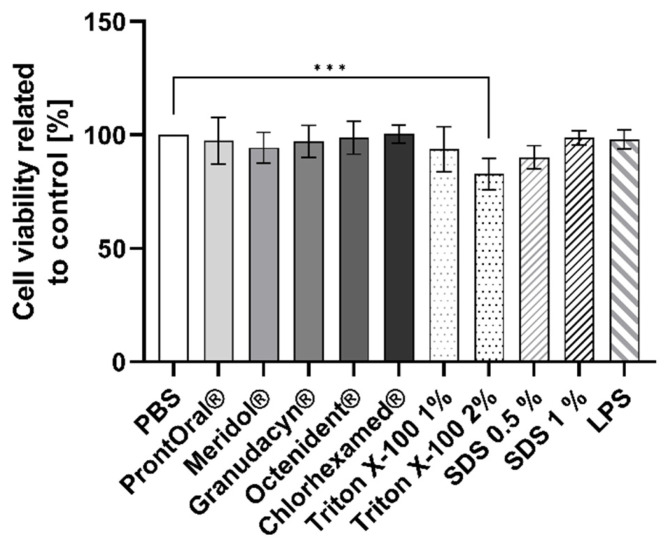
Cell viability (mean ± SD) relative to the control 24 h after treatment of the cells with the test products for 30 s (each *n* = 8–9; *** *p* < 0.001).

**Figure 6 pharmaceutics-15-00721-f006:**
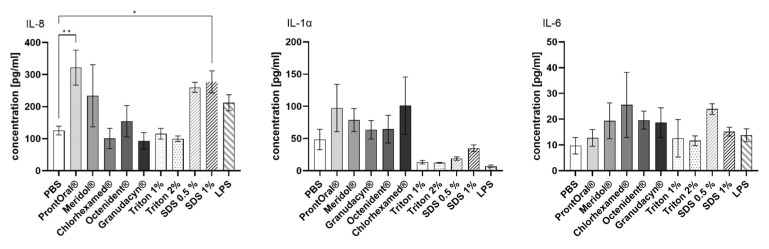
Amount (mean ± SEM) of secreted IL-8, IL-1α, and IL-6 24 h after 30-s treatment of the mucosa equivalents with the antiseptic rinses (each *n* = 4–8; * *p* < 0.05, ** *p* < 0.01).

## Data Availability

Data is contained within the article or are available from the corresponding author on request.

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
