# Peer review of "A 3D Model of Human Buccal Mucosa for Compatibility Testing of Mouth Rinsing Solutions"

_pharmaceutics, 2023, doi:10.3390/pharmaceutics15030721_

Round 1
Reviewer 1 Report
Thank you for a informative, relevant and interesting article
Author Response
We thank the reviewer for this positive feedback.

Reviewer 2 Report
The manuscript “A 3D model of human buccal mucosa for compatibility testing of mouth rinsing solutions” seems interesting and will be useful for oral mucosal drug delivery. However, I will suggest the following points before publication:
1) There are several reports, testing in vitro buccal drug delivery with TR146 cell layers. Could you discuss more in the introduction based on the recent publications?
2) Line 119 and onwards, is it good to call tissue (as these are cell layers)? Please confirm it.
3) The developed 3D might be compatible with testing oral rines. Do this model good for drug permeability studies? Have you tested or have any data?
Author Response
The manuscript “A 3D model of human buccal mucosa for compatibility testing of mouth rinsing solutions” seems interesting and will be useful for oral mucosal drug delivery. However, I will suggest the following points before publication:
1) There are several reports, testing in vitro buccal drug delivery with TR146 cell layers. Could you discuss more in the introduction based on the recent publications?
- Indeed, various publications are available testing buccal drug delivery with TR146 cell. However, the intention of our publication was to establish a 3D model that is suitable for compatibility testing of mouth rinses according to current OECD guidelines for epidermis models. That is why attention is paid on the tissue like structure, cytokeratin expression and TEER as well as resistance to chemicals and a high reproducibility. After confirmation of these properties, we used the model for a first test of cytocompatibility towards mouth rinses
- Drug delivery is not a main part of our manuscript. Nevertheless, we added some current publications in the introduction part (p 3, line 69), since testing of drug delivery might be conceivable for further research using our 3D model. Furthermore, we added some limitations of the study to the discussion part highlighting the use of drug delivery and permeability studies (p. 11, lines 394 ff.).
2) Line 119 and onwards, is it good to call tissue (as these are cell layers)? Please confirm it.
- The reviewer is right; the 3D cell model has a tissue like structure but is no “real” tissue. Thus, we replaced “tissue sections” by “cryo sections”.
3) The developed 3D might be compatible with testing oral rines. Do this model good for drug permeability studies? Have you tested or have any data?
- Thank you for this interesting question. Indeed, investigation of drug permeability delivers interesting data giving insights for drug delivery as the buccal mucosa is an alternative route for drug administration. Unfortunately, we did not test the model for use in drug permeability studies. However, these investigations will be conducted in further projects since our model seems to be promising for permeability and drug delivery studies (p. 11, lines 394 ff.):
Our study has some limitations. First, because we did not examine the permeability of the models, we cannot draw conclusions regarding the suitability for drug delivery testing nor do we have any data regarding delivery of antiseptic agents to deeper cell layers.

Reviewer 3 Report
Overall Conclusion:
1. Originality of the manuscript: Data original, but the concept is not new
2. Scientific merit: Average
3. Organization: Average
4. Clarity: Average
5. Adequate support of the conclusions: Yes
Minor Revisions * English language needs some revision and I advise the authors to seek help and advice in this regard. * In addition the authors should include in their discussion some explanation of what are the limitations of the study. My view of the manuscript is that it is an averagely written manuscript and contributing new knowledge in the subject. Therefore, authors should remain concise and to the point in delivering their data. This means the introduction and discussion sections can be revised. The discussion section also could be more streamlined to make its points more crisply. Grammatical corrections are clearly needed.
Author Response
English language needs some revision and I advise the authors to seek help and advice in this regard.
- Thank you for this advice. The manuscript was now corrected by a native speaker.
In addition the authors should include in their discussion some explanation of what are the limitations of the study.
Of course, our study has some limitations that we now explain in the discussion part (p. 11, lines 394 ff.).
- TR146 cells are a cancer cell line and thus they express a distinct phenotype in comparison to healthy cells. However, this limitation is present for a huge amount of in vitro studies dealing with cell lines, since cancer cell lines make up the main part of available cell lines. Furthermore, healthy cells isolated from patients have the disadvantage of high diversity and short cultivation periods, making them not useful for establishing 3D models.
- Furthermore, we did not have the possibility of comparison to excised human buccal mucosa.
- Another point is that we did not test the permeability of the cell model. These data would have given us more information about drug delivery to the cells and would have make it possible to explain reactions to the mouth rinses in more detail.
Our study has some limitations. First, because we did not examine the permeability of the models, we cannot draw conclusions regarding the suitability for drug delivery testing nor do we have any data regarding delivery of antiseptic agents to deeper cell layers. Furthermore, our study lacks a comparison to human buccal mucosa, since we did not examine our tests on excised mucosa. Finally, owing to the fact that we intended to establish a reproducible model, we used a cancer cell line that has a distinct phenotype in comparison to healthy cells, especially affecting wound healing and viability due to strengthened cell growth and reproduction. Because the 3D model closely resembles the situation of the buccal mucosa, it would be worthwhile to investigate its suitability for drug delivery testing.
My view of the manuscript is that it is an averagely written manuscript and contributing new knowledge in the subject. Therefore, authors should remain concise and to the point in delivering their data. This means the introduction and discussion sections can be revised. The discussion section also could be more streamlined to make its points more crisply. Grammatical corrections are clearly needed.
- Thank you for your advices. We shortened the introduction and the discussion to get more to the heart of our research.
- Furthermore, a native speaker corrected the manuscript.

Round 2
Reviewer 2 Report
The authors have edited the manuscript as per comments and suggestions. Thus, I recommend it for publication.